# Unveiling the Differences in Signaling and Regulatory Mechanisms between Dopamine D_2_ and D_3_ Receptors and Their Impact on Behavioral Sensitization

**DOI:** 10.3390/ijms24076742

**Published:** 2023-04-04

**Authors:** Kyeong-Man Kim

**Affiliations:** Department of Pharmacology, College of Pharmacy, Chonnam National University, Gwang-Ju 61186, Republic of Korea; kmkim@jnu.ac.kr

**Keywords:** dopamine D_2_ and D_3_ receptor, signal transduction, desensitization, intracellular trafficking, biased signaling, posttranslational modification, dimerization

## Abstract

Dopamine receptors are classified into five subtypes, with D_2_R and D_3_R playing a crucial role in regulating mood, motivation, reward, and movement. Whereas D_2_R are distributed widely across the brain, including regions responsible for motor functions, D_3_R are primarily found in specific areas related to cognitive and emotional functions, such as the nucleus accumbens, limbic system, and prefrontal cortex. Despite their high sequence homology and similar signaling pathways, D_2_R and D_3_R have distinct regulatory properties involving desensitization, endocytosis, posttranslational modification, and interactions with other cellular components. In vivo, D_3_R is closely associated with behavioral sensitization, which leads to increased dopaminergic responses. Behavioral sensitization is believed to result from D_3_R desensitization, which removes the inhibitory effect of D_3_R on related behaviors. Whereas D_2_R maintains continuous signal transduction through agonist-induced receptor phosphorylation, arrestin recruitment, and endocytosis, which recycle and resensitize desensitized receptors, D_3_R rarely undergoes agonist-induced endocytosis and instead is desensitized after repeated agonist exposure. In addition, D_3_R undergoes more extensive posttranslational modifications, such as glycosylation and palmitoylation, which are needed for its desensitization. Overall, a series of biochemical settings more closely related to D_3_R could be linked to D_3_R-mediated behavioral sensitization.

## 1. Introduction

Dopamine (DA), an ethylamine with an attached catechol group (a phenyl group with two hydroxyl groups attached to meta and para positions) is released from nerve endings upon axonal stimulation. Released DA acts on postsynaptic and presynaptic receptors at the synapse and is mostly taken up back into nerve endings by the dopamine transporter protein, which belongs to solute carrier transporter family.

DA modulates the neuronal activities in the brain regions that are innervated by three major dopaminergic pathways: nigrostriatal, mesocorticolimbic, and tuberoinfundibular. Drugs acting on D_2_R and D_3_R have traditionally been utilized for the treatment of various disorders including Parkinson’s disease [1], schizophrenia [2], and hyperprolactinemia [3].

The dopamine receptors are pharmacologically classified into D_1_-like and D_2_-like families [4,5,6]. D_1_-like receptors comprise D_1_ and D_5_ receptors (D_1_R, D_5_R) [7,8,9], whereas D_2_-like receptors consist of D_2_, D_3_, and D_4_ receptors (D_2_R, D_3_R, D_4_R) [10,11,12]. D_1_R and D_5_R mediate the stimulation of adenylyl cyclase by coupling to G_s_, whereas D_2_R, D_3_R, and D_4_R exert inhibitory effects on this enzyme by coupling to G_i/o_. D_2_-like receptors have larger third cytoplasmic loops and smaller carboxyl tails compared with D_1_-like receptors.

The architectures of D_2_R and D_3_R are similar, with 46% overall amino acid homology and 78% identity in the transmembrane domains [13]. Similarly, D_2_R and D_3_R share many signaling properties when they are expressed in mammalian cells. For example, both receptors inhibit adenylyl cyclase [14,15,16] and stimulate extracellular acidification (Na^+^/H^+^ exchange) [14,17,18], mitogenesis [14,19], MAPK activation [20,21], dopamine release [22], and the opening of inward rectifier potassium channels [23,24].

Whereas D_2_R and D_3_R exhibit a high degree of similarity in sequence homology and signaling properties, they differ in some other respects. For example, the transcript levels of D_3_R are much lower than those of D_2_R, differing by several orders of magnitude in their expression levels [25]. However, D_3_R shows about 150 times higher affinity for DA [26]. In addition, the distribution of D_3_R in the brain is more restricted than that of D_2_R, that is, D_3_R shows preferential expression in brain regions responsible for emotional and cognitive functions, such as the nucleus accumbens and the islands of Calleja and has been suggested as a therapeutic target for the affective and mood-associated diseases [27,28,29]. Moreover, significant variations exist between D_2_R and D_3_R with respect to their endocytosis, desensitization, and posttranslational modifications [30,31,32].

The main objective of this review is to comprehend the functional implications of the distinct regulatory processes of D_2_R and D_3_R by highlighting their differences. Functional in vivo roles and ligands of D_2_R and D_3_R are described in other excellent reviews [33,34,35,36,37]. This review focuses on the regulatory functions of D_2_R and D_3_R in biochemical pathways.

## 2. Desensitization and Intracellular Trafficking

### 2.1. Desensitization

Desensitization of G protein-coupled receptors (GPCRs) represents a gradual attenuation of receptor responsiveness by continuous or repeated exposure to agonists [38,39,40]. GPCR desensitization is generally classified into two categories in which either GRKs/arrestins or second messenger-regulated kinases are involved. The former desensitizes only agonist-activated receptors, whereas the latter can desensitize receptors both in an activation-dependent and activation-independent manner [40]. Typical second messenger-regulated kinases are protein kinase A (PKA) and protein kinase C (PKC) [41,42,43].

The desensitization of GPCRs has been most extensively characterized by the conventional or steric-hindrance-based uncoupling model. When these receptors are activated by agonists, they undergo conformational changes that involve the outward movement of transmembrane helix VI. This results in the formation of a cavity that allows for the binding of G proteins [44,45,46]. The (E/D)x(I/L)xxxGL motif, which is commonly present in the C-termini of Gα and arrestins, is believed to participate in their interaction with the active GPCRs at the same binding crevice [47,48]. The binding of arrestins to GPCRs is likely to occur in competition with G proteins. By binding to the same binding crevice, arrestins can obstruct further coupling of G proteins via steric hindrance, which ultimately leads to a decrease in receptor signaling [49].

Even though D_2_R and D_3_R have similar structural features and signaling pathways, they display distinct properties of desensitization and intracellular trafficking. In response to agonistic stimulation, D_2_R is phosphorylated in a manner dependent on GRK2/3 and in part on GRK5 and mediates arrestin translocation to the plasma membrane accompanied by receptor endocytosis [30]. In contrast, D_3_R is rarely phosphorylated and undergoes only a minute amount of endocytosis.

According to the steric-hindrance-based uncoupling model, D_2_R but not D_3_R would show agonist-induced desensitization. However, subsequent studies have reported the desensitization of D_3_R but not D_2_R [50,51,52], suggesting that the steric-hindrance-based uncoupling model may not be applicable to the desensitization of D_2_R and D_3_R or desensitized D_2_R might be rapidly resensitized [51,53].

Recent studies conducted on D_2_-like receptors (D_2_R, D_3_R, D_4_R) and the β_2_ adrenoceptor (β_2_AR) indicate that receptors that undergo desensitization have a predisposition to form a stable complex with Gβγ and enable a basal interaction between Gβγ and arrestins [54]. This arrangement leads to the recruitment of Mdm2 to the cytoplasm by these receptors in their basal state, ultimately resulting in the constitutive ubiquitination of arrestins in the cytoplasm [55].

In a recent study, it was suggested that certain GPCRs, including D_3_R and β_2_AR, undergo desensitization via a cascade involving Src, PDK1, and Akt. Akt then interacts with USP33, an arrestin deubiquitinase, to facilitate the deubiquitination of arrestins [32]. Following deubiquitination, arrestins form a complex with Gβγ and translocate to the nucleus via an importin complex. Here, they sequester Gβγ from the receptor and Gα, thus dampening receptor signaling. Therefore, it is plausible that appropriate desensitization measures are in place to properly control the cellular responses depending on the characteristics of receptors, agonists, and other cellular factors.

In the case of D_2_R, which does not display noticeable desensitization under the same experimental conditions where D_3_R undergoes desensitization [50,52], GRK-mediated receptor phosphorylation is needed for the recycling of endocytosed receptors [53]. This recycling accompanies the dissociation of arrestins from the receptor, which is believed to be the cellular process required for the resensitization of D_2_R [51].

In contrast to agonist-induced desensitization, which is mainly associated with D_3_R, PKC-mediated desensitization affects both D_2_R and D_3_R [43,56].

The amino acid residues S145/146 and C147 are responsible for the agonist-induced desensitization of D_3_R [50,52]; S325 for the PKC-mediated desensitization of D_2S_R (short isoform of the D_2_R generated by alternative splicing) [56]; and S229 and S257 for the PKC-mediated desensitization of D_3_R [43] (Table 1, Figure 1 and Figure 2).

### 2.2. Endocytosis

The process of endocytosis involves the uptake of extracellular materials by cells through the inward budding of vesicles that originate from the plasma membrane [60]. Endocytosis of the receptor occurs without failure when agonist stimulation induces receptor phosphorylation and arrestin translocation. Translocated arrestins mediate receptor endocytosis [61,62] and additional components of endocytic machinery, such as dynamin and β2-adaptin, are also involved in this process [63,64]. The interaction of arrestins with receptors in endocytic vesicles is dependent on the affinity between the two, with varying levels of association observed [65]. Class A receptors, such as the β_2_AR and D_1_R, exhibit lower affinity towards arrestins compared with class B receptors. As a result, these receptors tend to dissociate from arrestins when they are incorporated into endocytic vesicles. In contrast, class B receptors, such as the angiotensin II type 1A receptor and vasopressin V2 receptor, maintain their interaction with arrestins during endocytosis. This ability of arrestins to remain associated with class B receptors is facilitated by specific clusters of serine and threonine residues located in the carboxyl-terminal tails of the receptors [66].

Similar to β_2_AR, D_2_R undergoes agonist-induced endocytosis. GRK2/3 and GRK5 in part mediate the phosphorylation of D_2_R. In contrast, GRK-mediated phosphorylation rarely occurs in D_3_R, which undergoes only a minute amount of endocytosis. It has been reported that S/T residues located in the second (T134, T144, S147, S148) and the third (T225) intracellular loops of D_2_R are involved in agonist-induced D_2_R endocytosis (Cho et al., 2010) (Figure 1). Another study has shown that some other S/T residues located within the third intracellular loop of D_2_R (S256, S257, T258, S259, T264, S282, S288, S292) are also phosphorylated in a GRK-mediated manner and that they are required for the agonist-induced (GRK-mediated) recycling of endocytosed D_2_R [53] (Table 1, Figure 1). It is interesting that different sets of S/T residues mediate distinct intracellular trafficking processes. The resensitization of D_2_R occurs when the receptor dissociates from arrestin via agonist-induced endocytosis [51].

In addition to GRK/arrestin-mediated regulation, D_2_R and D_3_R are also regulated by PKC. Robust phosphorylation of D_2_R is induced by PMA (phorbol 12-myristate 13-acetate) but the resulting endocytosis is weaker than the agonist-induced endocytosis of D_2_R [43,51]. The phosphorylation, endocytosis, and desensitization of D_3_R induced by PMA are more robust than those of D_2_R [43,51,56]. It has been reported that treatment with PMA results in the ubiquitination of D_3_R and its subsequent degradation through the lysosomal pathway. The degradation of D_3_R by PKC is dependent on clathrin-mediated endocytosis. However, this process is distinct from the desensitization of D_3_R and is not affected by its inhibition [59]. The D_2_R residues T225, S228, S229, T322, T324, and S325 are responsible for PMA-induced receptor phosphorylation and endocytosis [56,67] (Table 1, Figure 1). PKC-mediated phosphorylation and endocytosis of D_3_R are mediated by the S229 and S257 residues [43] (Table 1, Figure 2).

Even though D_3_R does not undergo typical endocytosis, which involves the movement of receptors from the plasma membrane to the cytosol, it undergoes another type of intracellular trafficking called pharmacological sequestration [52,68]. Upon desensitization of D_3_R induced by an adequate amount of agonist, it translocates to the hydrophobic region of the membrane where the hydrophilic agonist has reduced binding efficiency. The time course of D_3_R desensitization and pharmacological sequestration are similar, and both depend on the presence of Gβγ and arrestin, indicating that the phenomenon of pharmacological sequestration may account for D_3_R desensitization.

The movement of GFP-tagged receptors from the plasma membrane to the cytosol upon agonist treatment is a convenient way to confirm the endocytosis of receptor proteins. Alternatively, biochemical assays such as fluorescence-activated single cell sorting (FACS) or enzyme-linked immunosorbent assay (ELISA) can be employed using constructs with HA or FLAG epitopes attached to the N-terminus of the receptor. Hydrophilic radioligands can also be used to assess receptor endocytosis as they cannot bind receptors in the cytoplasm. However, caution should be exercised when dealing with receptors that have high affinity for the agonist, such as DA and D_3_R. In such cases, it is difficult to completely wash away the agonist from cells expressing D_3_R unless the cells are washed under harsh conditions, such as with a low pH buffer. If cells are treated with a radioligand under conditions where a pre-applied agonist is not completely washed, it may give a false impression that receptor endocytosis is occurring (approximately 30%) because the radioligand cannot properly bind to the receptor that is already occupied by the tightly bound agonist.

### 2.3. Roles of Small G Proteins in the Signaling and Endocytosis of D_2_R and D_3_R

There are two families of GTP binding proteins: heterotrimeric large G proteins composed of three subunits (α, β, and γ) and small G proteins with a single subunit. Small G proteins mediate more versatile functions than trimeric G proteins due to their larger number in the cell [69,70]. Small GTPases can be categorized into five main families, namely Ras, Rho, Rab, ADP-ribosylation factor (Arf), and Ran, based on both sequence and functional criteria [70].

Small GTPases, including Rab and ARF proteins, as well as the large GTPase dynamin, play regulatory roles in vesicular transport [71]. Several Rab proteins, including Rab5 and Rab23, are involved in the regulation of membrane trafficking and recycling of cell surface proteins from the Golgi to the plasma membrane [72,73].

Whereas RalA and ARF6 small G proteins are known to regulate D_2_R and D_3_R [67,74,75], the involvement of Rab5 and Rab23 in the endocytosis of D_2_R remains controversial [67,76].

#### 2.3.1. RalA

RalA belongs to the Ras family of small G proteins. The GTP-bound form (G23V, active) of RalA is known to inhibit the endocytosis of D_2_R in a manner that is independent of the previously reported downstream effectors of RalA, such as Ral-binding protein 1 and phospholipase D [74]. The endocytosis of GPCRs, including D_2_R, can be inhibited by active RalA through its high affinity for GRK2, which sequesters GRK2 away from the GPCRs. RalA is converted from its active to inactive state by the translocation of Ral GDP dissociation stimulator-like (RGL), a guanine nucleotide exchange factor, from the plasma membrane to the cytosol in a complex with Gβγ. Thus, the agonist-induced Gβγ-mediated conversion of RalA from its GTP-bound to GDP-bound form is thought to trigger and facilitate the endocytosis of GPCRs induced by agonist stimulation [74].

Filamin A (FLNA), an actin-binding protein, functions as a signaling and intracellular trafficking scaffold for various GPCRs, including D_2_R and D_3_R. FLNA plays an essential role in maintaining the proper expression of D_2_R on the plasma membrane [77] and is also necessary for signaling through both D_2_R and D_3_R [78,79,80]. FLNA is known to constitutively inhibit RalA activity, thus ensuring proper receptor trafficking and signaling. In the absence of FLNA, RalA is activated and sequesters GRK2 from the receptor, resulting in desensitization of D_2_R due to its inhibition of recycling [51,53]. On the other hand, active RalA inhibits the signaling of D_3_R, which does not undergo agonist-induced endocytosis, in an arrestin-dependent manner. Thus, it is suggested that active RalA modulates receptor signaling differently depending on whether GRK2 or arrestin is involved in the functional regulation of the respective receptor.

#### 2.3.2. ARF6

The ARF (ADP-ribosylation factor) family of small GTPases comprises six members, among which ARF6 is the most extensively studied with regards to its role in intracellular trafficking of membrane proteins. ARF6 is known to accumulate in clathrin-coated pits (CCPs) in a GTP-dependent manner and facilitate the rapid recycling of plasma membrane receptors [81,82]. Furthermore, research suggests that the activation of ARF6 via arrestin-mediated mechanisms is involved in the endocytosis of β_2_AR [83].

According to a report, the endocytosis of D_2_R is not influenced by ARF6, as neither constitutively active nor a dominant-negative mutant of ARF6 has any effect [67]. However, the recycling of endocytosed D_2_R is inhibited by a constitutively GTP-bound mutant of ARF6 (Q67L), indicating that the conversion of ARF6 from the GTP-bound to GDP-bound form is crucial for the proper recycling of endocytosed vesicles.

### 2.4. Roles of Regulators of G Protein Signaling (RGS) in the Signaling of D_2_R and D_3_R

In its inactive state, the Gα subunit is bound to GDP, whereas activation of the GPCR by an agonist triggers the exchange of GDP for GTP on the Gα subunit, resulting in its dissociation from the Gβγ heterodimer [84]. Both the Gα and Gβγ subunits are capable of regulating downstream effector proteins. The Gα subunit possesses intrinsic GTPase activity, which converts the GTP-bound Gα to its inactive form, Gα-GDP, allowing it to re-associate with the Gβγ heterodimers and effectively terminate signaling via both Gα-GTP and Gβγ.

The rapid physiological timing of GPCR signal transduction observed in vivo contrasts with the slow rates of GTP hydrolysis exhibited by purified Gα subunits in vitro. This discrepancy has been explained by the existence of the GTPase-accelerating protein (GAP) family [85,86]. Unlike the five main families of GAPs that regulate the Ras superfamily, regulators of G protein signaling (RGSs) specifically target heterotrimeric G proteins [87]. RGS proteins bind to and regulate the Gαi and Gαq subfamilies of proteins by increasing their GTPase activity. However, the Gαs subfamily already exhibits a sufficiently high intrinsic GTPase activity and is not sensitive to RGS protein-mediated GAP activity.

RGS2, RGS4, and RGS9-2 are RGS subtypes that have been associated with the dopaminergic nervous system. In a previous study utilizing in situ hybridization, it was suggested that D_1_R and D_2_R co-localize with RGS2 and RGS4, respectively [88]. It has also been shown that D_1_R and D_2_R regulate the expression of RGS2 and RGS4 in opposite directions. For instance, quinpirole, a D_2_-like receptor agonist, induces the downregulation of RGS2 and upregulation of RGS4 [89]. Interestingly, male rats require RGS4, but not RGS2, for the rewarding effects of cocaine [90]. Further studies have demonstrated that the N-terminal domain of RGS4 interacts with D_2_R and D_3_R and inhibits receptor signaling via the RGS domain [57].

RGS9-2 exhibits high levels of expression in the striatum and nucleus accumbens, regions that also express D_2_-like receptors. Studies suggest that RGS9-2 plays a functional role in regulating D_2_-like receptors. Specifically, introducing RGS9-2 through viral expression in brain regions such as the nucleus accumbens or introducing RGS9-2 proteins into striatal cholinergic interneurons reduces the behavioral or electrophysiological response to D_2_-like receptor stimulation [91,92,93]. In contrast, when RGS9-2 is knocked out, there is an augmentation in the behavioral response to D_2_-like receptor activation [91,93].

RGS9-2 is a member of the R7 RGS subfamily and possesses three domains: RGS, disheveled-EGL10-pleckstrin (DEP) homology, and G-gamma-like (GGL). The RGS domain binds to the Gα subunit and facilitates GAP activity. The DEP domain anchors the protein to the membrane and allows for interaction with the C-terminal tail of GPCRs. Finally, the GGL domain ensures protein stability by dimerizing with Gβ5 [94,95,96,97].

In HEK-293 cells or C6 glioma cells, the expression of RGS9-2 specifically hinders D_3_R signaling while leaving D_2_R/D_4_R signaling unaffected. This can be attributed to the varying affinities of the receptors for arrestin3, which facilitates the creation of a complex consisting of RGS9’s DEP domain, Gβ5, R7-binding protein (R7BP), and D_3_R [98].

## 3. Biased Signaling

GPCRs carry out their functions through various signaling pathways, which can be categorized into G protein-dependent and G protein-independent pathways (Wisler et al., 2018). The G protein-independent pathways mainly involve arrestins, which were previously believed to be involved in agonist-triggered receptor desensitization and endocytosis [61,99,100]. Unlike conventional balanced agonists that stimulate both G protein and arrestin pathways, newly synthesized ligands have been reported to selectively act on one of the two pathways [101,102,103]. It is speculated that biased agonists stabilize GPCRs in a specific conformation [104,105].

There has been great interest in the development of biased ligands because they are proposed to be superior to traditional balanced agonists in terms of efficacy and adverse effects in the treatment of certain disorders [106,107]. Usually, biased ligands have been designed based on data obtained through structure–activity analyses of currently available ligands [35,108]. Biased signaling can occur not only in response to tailored ligands but also through genetically modified GPCRs, which can provide new insights for designing novel biased ligands. Several biased D_2_Rs have been reported, and some of them are discussed below (Figure 1, Table 2).

Structure–activity analyses of D_2_R agonists revealed that a hydrophobic pocket at the interface of the second extracellular loop and fifth transmembrane segment of D_2_R is involved in biased signaling [108]. One of the point mutations examined in D_2_R, specifically the mutation at F189, which is equivalent to F^5.38^ according to Ballesteros–Weinstein GPCR numbering system [112], showed a preference for G protein coupling. When tested for G protein signaling, the DA potency for F189A-D_2_R decreased about 500 times compared with that for WT-D_2_R; however, the interaction with arrestin2 was almost completely abolished, resulting in impaired arrestin recruitment while maintaining G protein signaling to some extent.

The Asp–Arg–Tyr (DRY) motif is located at the beginning of the second intracellular loop. The Arg residue within this motif plays vital roles in G protein coupling, receptor phosphorylation, and arrestin recruitment by forming intramolecular interactions [113,114]. To create a biased D_2_R, a point mutation was introduced at R132 in the DRY motif (R^3.50^), so that both G protein coupling and arrestin translocation were suppressed. An additional mutation was then introduced at L^3.41^ (L123W), which determines the thermostability [115], to increase the potency for DA [109]. The resulting D_2_R mutant (R132L/L123W) produced displayed biased signaling through G proteins. This is because the L123W mutation, in addition to R132L, significantly restored the signaling ability of R132L-D_2_R through G proteins while having a lesser impact on its ability to recruit arrestins.

Research also revealed that modifying four amino acid residues within the N-terminal region of the third intracellular loop (^212^IYIV^215^) resulted in a signaling-biased receptor. The receptor maintained its ability to bind ligands and couple with and activate G proteins but demonstrated impaired ability to mediate arrestin3 translocation to the plasma membrane [110].

Using the evolutionary trace (ET) method [116], another study aimed to develop a biased D_2_R. Peterson et al. successfully identified D_2_R mutants that exhibited selective signaling through either G protein or arrestins [111,117]. The team discovered two D_2_R mutants with dual amino acid substitutions, L125N/Y133L and A135R/M140D, which showed a preference for G protein-mediated and arrestin-dependent pathways, respectively. These modified D_2_R mutants were then used for in vivo studies to investigate the roles of the arrestin pathway in amphetamine-induced locomotion potentiation [117].

Identifying the sites that determine biased signaling in D_3_R might not be straightforward, as agonist treatment rarely induces arrestin recruitment [30]. To measure arrestin-dependent signaling, the carboxyl terminus of the vasopressin type 2 receptor is typically attached to the receptor for stable arrestin translocation [118]. In the case of D_2_R, this approach leads to a complementary interaction with the DRY motif, whereas in the case of D_3_R it causes a shift in the arrestin translocation pattern from type A to type B [113,114]. In type A GPCRs, arrestin is released from the receptor as endocytosis progresses, whereas in type B GPCRs arrestin remains bound to the receptor and accompanies it in the endocytic vesicle [65].

A previous study has demonstrated that a G protein-biased ligand interacts with Asp110 on transmembrane 3 and His349 on transmembrane 6, but it is unclear whether these amino acids are the decisive factors for the biased signaling of D_3_R [119]. Although the involvement of the arrestin-dependent pathway has been suggested in D_3_R signaling, identifying the amino acid residues responsible for biased signaling and creating new biased agonists will require both conceptual and methodological advancements [120].

A previous study showed that the knockdown of arrestins does not affect D_2_R-mediated ERK activation [121], which is contradictory to recent publications indicating the existence of biased mutant D_2_Rs and ligands [35,108,109,110,111,122,123].

There are several potential reasons for the discrepancy observed in the effects of arrestin depletion on ERK activation. One possible explanation is that the G protein and arrestin pathways are complementary for ERK activation, meaning that the remaining pathway can compensate for the absence of the other. Another possibility is the dual role of arrestins in mediating both their own pathway, such as ERK activation, and desensitizing the G protein pathway. Therefore, removing endogenous arrestin may block arrestin-dependent ERK activation but enhance G protein-dependent ERK activation. Another consideration is the technical limitations of siRNA- or shRNA-mediated knockdown, which may not completely remove the target protein. If a small amount of arrestin is sufficient to support the arrestin signaling pathway, ERK activation may not be affected even with significant arrestin depletion.

The identification of the factors that determine ligand binding and understanding the 3D structure of D_2_R and D_3_R can greatly aid in developing selective or biased ligands for these receptors. Specifically, the structural analyses of D_2_R or D_3_R in complex with G protein or arrestin can have significant implications for developing biased ligands. Recent research has revealed the crystal structures of D_2_R bound to risperidone and D_3_R bound to eticlopride [124,125], as well as the 3D structures of D_2_R or D_3_R in complex with Gi through cryo-electron microscopy analysis [126,127]. Additional studies linking computational techniques with experimental results have identified factors involved in ligand binding, further advancing structure-based drug design [128,129]. Together, these findings enhance our understanding of the structural and functional properties of D_2_R and D_3_R and underscore the potential of structure-based drug design to develop novel therapeutics targeting these receptors.

## 4. Dimerization of Dopamine D_2_ and D_3_ Receptors

GPCRs have the ability to form dimers or oligomers with similar or different types of GPCRs. Coactivation of D_1_R and D_2_R, which couple to Gs and Gi/o, respectively, has been demonstrated to activate Gq protein, leading to intracellular calcium release through the activation of phospholipase C and IP3 [130,131,132].

The D_1_R and D_2_R interact through adjacent glutamic acids in the carboxyl tail of D_1_R (404E/405E) and two adjacent arginine residues in the third intracellular loop of D_2_R (245R/246R) [133]. Previously, the third intracellular loop of D_2_R, specifically the region ^217^RRRRKR^222^, was identified as a potential site for heteromer interaction, forming heteromers with D_1_R [134], 5-HT_2A_R [135], or adenosine A2A receptors [136].

Studies have demonstrated both direct and functional interactions between D_1_R and D_3_R. Depending on the cell type or signaling, the two receptor subtypes may affect neurons in either a synergistic or opposing manner [137,138,139]. Furthermore, D_3_R stimulation has been shown to enhance D_1_R agonist affinity, which potentiates D_1_R-mediated behavioral effects [140,141].

Dimerization between these two receptors has also been observed to alter endocytic properties. Heterodimerization with D_3_R, for instance, abolishes agonist-induced endocytosis of D_1_R but allows the endocytosis of the D_1_R/D_3_R complex in response to simultaneous agonistic stimulation of both receptors [141]. It is noteworthy that G protein-independent signaling can occur with stimulation of D_1_R/D_3_R heteromers in the nucleus accumbens [142].

There have been several studies indicating the functional interactions between D_3_R and nAChR in vivo. For instance, D_3_R and α4β2 nAChR form heteromers in dopaminergic neurons that are crucial for the neurotrophic effects of nicotine [143]. Furthermore, the development of nicotine sensitization is accompanied by an increase in the expression of D_3_R [144,145], and it has been suggested that D_3_R ligands can be used to treat tobacco dependence [146]. A recent in vitro study showed that nicotinic stimulation of α4β2 nAChR leads to Src activation in an arrestin2- and 14-3-3η-dependent manner. The activated Src phosphorylates the tyrosine residue(s) on Syk molecules that then interact with phospholipase Cγ1 to trigger the translocation of PKCβII to the cell membrane by elevating cellular diacylglycerol levels [147].

## 5. Posttranslational Modifications

Posttranslational modifications (PTMs) are chemical modifications of amino acid side chains that occur after protein synthesis is complete. These modifications usually take place in the endoplasmic reticulum (ER) and the Golgi apparatus [148]. In the case of GPCRs, PTMs can occur in all regions of the receptor except for the transmembrane domains. These modifications are important for regulating receptor folding, maturation, trafficking, and signaling, which increases functional diversity and fine-tunes signaling pathways [149,150]. The most extensively characterized PTMs include glycosylation, phosphorylation, ubiquitination, and palmitoylation [150,151]. The sites and functional roles of PTMs of D_2_R and D_3_R are summarized in Table 1 and Figure 3.

### 5.1. Glycosylation

Glycosylation takes place prevalently at the N-termini or extracellular loops of GPCRs. *N*-glycosylation, the major form of glycosylation in GPCRs, links a sugar molecule to the nitrogen of the Asn (N) residue in the consensus motif N-X-S/T (X ≠ P) [152,153]. Detection of *N*-glycosylation in target proteins has traditionally involved enzymatic cleavage with peptide *N*-glycosidase F (PNGase F) or endoglycosidase H (Endo H) along with site-directed mutagenesis in the consensus sequence.

The process of glycosylation commences with dolichol, which is a polymerized isoprene molecule. In the ER membrane, dolichol, with its hydroxyl group facing the cytoplasmic side, is converted to dolichol phosphate [154]. Subsequently, *N*-acetylglucosamine (NAG)-UDP and β-D-mannopyranose are added in succession to dolichol phosphate. Dolichol then flips so that the entire glycosyl moiety faces the ER lumen and additional mannose (M) moieties along with glucose (G) molecules are added to form a large complex of dolichol diphosphate–NAG-NAG-M/Gs, which is now ready for the glycosylation of the target protein. Glycosylation takes place when the NAG-NAG-M/G moiety, detached from dolichol diphosphate, interacts with the nitrogen atom on the side chain of the asparagine residue in the target protein.

Glycosylation is involved in the modulation of various aspects of receptor function, such as maturation, trafficking, ligand binding, and cell signaling, as reported in several studies [155,156,157]. The D_2_R possesses three *N*-linked glycosylation sites at its N-terminus (N5, N17, N23), whereas the D_3_R has four *N*-linked glycosylation sites. Two of these sites are located at the N-terminus (N12, N19), whereas the other two are present in the second and third extracellular loops (N97 and N173, respectively).

Surface expression of both D_2_R and D_3_R is commonly affected by glycosylation at the N-terminus, with D_2_R undergoing caveolae endocytosis and D_3_R undergoing clathrin-mediated endocytosis. In the case of D_3_R, glycosylation at the N-terminus is necessary for basal signaling and desensitization, whereas glycosylation at N97 and N173 in the second and third extracellular domains, respectively, is involved in receptor endocytosis [57,58,158].

### 5.2. Phosphorylation

Following agonist binding, GPCRs undergo conformational changes and interact with a specific heterotrimeric G protein, which enhances its GTPase activity, resulting in the separation of the G protein into α and βγ subunits that subsequently modulate downstream signaling pathways [84]. Concurrently, regulatory processes are initiated following receptor activation, with receptor phosphorylation playing a crucial role [159,160].

The intracellular regions of GPCRs contain numerous phosphorylation sites that are predominantly targeted by two classes of serine/threonine kinases: GPCR kinases (GRKs) and second-messenger-dependent protein kinases, such as protein kinase A (PKA) and protein kinase C (PKC) [40,42,43,161]. Techniques such as mass spectrometry, phosphor-specific immunoblotting, and site-directed mutagenesis of the consensus sites in the target proteins have been utilized to identify GPCR phosphorylation sites [162,163].

Protein kinases, including PKA and PKC, recognize specific substrate sequences independent of receptor activation [49,164]. Furthermore, the activation of these kinases is reliant on second messengers that can spread throughout the cell [42,43,165]. As a result, it is likely that both occupied and unoccupied neighboring receptors are phosphorylated by these kinases.

GRKs have a preference for phosphorylating GPCRs when the receptors are in an active state, which means that they are occupied by agonists [166]. This selectivity is due to the dynamic interactions between GRKs and GPCRs, as both proteins undergo conformational changes upon agonist stimulation of GPCRs [167,168]. The interaction and activation of GRKs are mainly mediated by the overall topologic structure of the activated receptor rather than the amino acid sequence surrounding the phosphorylation site [167,169,170,171]. Activated receptors also enhance the enzymatic activity of GRKs, making them both a substrate and activator of GRKs [167,170,172].

The intracellular regions of D_2_R and D_3_R contain a number of serine and threonine residues. The short isoform of D_2_R (D_2S_R) has three threonine residues located in the first intracellular loop, two serine and two threonine residues in the second loop, and a total of thirteen serine and ten threonine residues in the third loop. On the other hand, D_3_R includes three threonine residues in the first intracellular loop, two serine and two threonine residues in the second loop, ten serine and seven threonine residues in the third loop, and an additional serine residue in the carboxyl terminal region.

Phosphorylation of D_2_R is induced by agonists at various serine and threonine residues, including S256, S257, T258, S259, T264, S282, S288, and S292 [53]. PKC-mediated phosphorylation of D_2S_R, which is inferred from D_2L_R results, mainly occurs at residues 228, 229, and 325, with additional contributions of threonine residues at 322 and 324 [56].

D_3_R is phosphorylated at S229 and S259 upon PMA stimulation, and these phosphorylations play a significant role in regulating desensitization and intracellular trafficking of the receptor [43].

### 5.3. Ubiquitination

Ubiquitin is a polypeptide consisting of 76 amino acids that is found in eukaryotes. The process of ubiquitylation involves the formation of an isopeptide bond that connects an internal lysine residue of the target protein with the C-terminal glycine (glycine 76) of ubiquitin (Hershko, 2005). Ubiquitination requires three sets of enzymes: Ub-activating (E1), Ub-conjugating (E2), and Ub-ligating enzymes (E3) [173]. E1 and E2 sequentially form thioester bonds with ubiquitin, which involves condensation and conjugation between the thiol groups (usually of cysteine) of E1 and E2 and the carboxyl group of Ub (glycine 76). E2 then associates with an E3, through which ubiquitin is transferred from E2 to E3. Finally, E3 catalyzes the conjugation between the amino group of a lysine residue on the target substrate and the carboxyl group of Ub via an isopeptide bond. E3 plays a crucial role in ubiquitination by conferring substrate specificity [174].

There are three different ways in which substrate proteins can be attached to Ub, namely monoubiquitination, multi-monoubiquitination, and polyubiquitination. To form polyubiquitin chains, eight residues in ubiquitin can be employed, including K6, K11, K27, K29, K33, K48, K63, and M1 (linear). Depending on the type of linkage, the modified proteins are directed towards diverse cellular outcomes [175,176].

Regulating biosynthesis, endocytosis, lysosomal sorting, degradation, and cellular signaling of GPCRs heavily rely on ubiquitination [150,177]. Typically, intracellular lysine residues can undergo either monoubiquitination or polyubiquitination in response to agonist-dependent or agonist-independent stimuli [178,179].

Although PMA treatment has been found to induce ubiquitination and lysosomal degradation of D_3_R [59], there is no evidence of agonist-induced ubiquitination of D_2_R and D_3_R.

### 5.4. Palmitoylation

The initial step of palmitoylation involves creating a thioester linkage between the carboxyl group of palmitic acid (hexadecanoic acid, C16) and the thiol group of coenzyme A (CoA). *S*-palmitoylation, which is more frequent than *N*-palmitoylation, involves the thiol group of a serine residue in the target protein attacking the carbonyl group of the thioester linkage formed in the first step [180]. This results in the formation of a fresh thioester linkage between the target protein and palmitic acid. Meanwhile, CoA is released and can be reused. The precise consensus sequences surrounding the cysteinyl residues that undergo palmitoylation have yet to be determined [181].

When methionine, the first amino acid of a polypeptide, is removed, *N*-palmitoylation occurs when the first amino acid is either cysteine, glycine, or lysine [182]. Similar to *S*-palmitoylation, the free amino group of these amino acids in the target protein reacts with the carbonyl group of the thioester linkage established in the first step. As a result, a new amide linkage is formed between the target protein and palmitic acid.

*S*-palmitoylation can occur spontaneously or can be catalyzed by multi-span transmembrane integral proteins known as protein acyl transferases (PATs). PATs possess zinc-finger and aspartate–histidine–histidine–cysteine (zDHHC) domains [183]. The intracellular DHHC motif plays a crucial role in the *S*-acylating activity of PATs, allowing protein palmitoylation to occur at the cytoplasmic face of membranes in the secretory pathway, including the ER and Golgi apparatus as well as the plasma membrane.

Palmitoylation is a reversible process, and depalmitoylation is mediated by protein thioesterases or depalmitoylases. Currently, three classes of depalmitoylases have been identified; acyl-protein thioesterases (APTs), α/β hydrolase domain-containing 17 proteins (ABHD17s), and palmitoyl-protein thioesterases (PPTs) [184]. APTs located in the cytosol are responsible for depalmitoylating targets of Gα and Ras [185,186]. In contrast, PPTs found in the lysosome play a role in regulating target protein degradation [187].

During the biosynthesis of GPCRs, palmitoylation often occurs at their C-termini and sometimes at intracellular loops. This modification is usually a basal process but can also be induced by agonist stimulation in some cases [188]. Palmitoylation is essential for various functions of GPCRs, including trafficking, cell surface localization, dimerization, and signaling [188,189].

D_2_R and D_3_R share a conserved cysteine residue at their carboxyl termini and have a highly similar sequence for the last 10 amino acid residues except for the second-to-last residue (histidine in D_2_R and serine in D_3_R). Despite this similarity, D_3_R is constitutively palmitoylated at C400, whereas D_2S_R is not palmitoylated in HEK-293 cells [31]. Palmitoylation is a critical modification for proper localization of D_3_R to the cell surface, as well as for PKC-mediated endocytosis, agonist affinity, and desensitization of the receptor [31].

Studies have shown that the palmitoylation of D_2_R may vary depending on the expression system. D_2S_R was not palmitoylated in HEK-293 cells [31], whereas other studies have reported that both D_2S_R and D_2L_R are palmitoylated when expressed in the baculovirus/Sf9 cell system [190,191]. Moreover, conflicting results have been reported on the palmitoylation site of D_2_R, with one study suggesting C443 as the palmitoylation site [192] and another study reporting no palmitoylation at this site [31]. These discrepancies may have resulted from differences in experimental strategies and conditions, such as differences in the concentration and duration of 2-bromopalmitate treatment and the use of deletion or point mutation strategies to disrupt palmitoylation.

Taking into account the variations in experimental conditions, it is probable that only D_3_R (not D_2_R) undergoes palmitoylation at the C-terminus. Furthermore, although palmitoylation may occur in D_2_R, it appears that D_3_R is more heavily palmitoylated. Additionally, palmitoylation seems to have a more crucial impact on the regulation of D_3_R’s functions than those of D_2_R.

Although the consensus sequence for palmitoylation is not yet fully established, a comparison between D_2_R and D_3_R yielded some insight. Specifically, the presence of a Ser residue preceding the Cys residue appears to be crucial for palmitoylation of D_3_R. For instance, when the C-terminal “HC” sequence of D_2_R, which is typically not palmitoylated, is modified to “SC” as in D_3_R, D_2_R becomes palmitoylated, and the reverse is also true [31].

## 6. Conclusions

Establishing a correlation between the in vivo roles of certain GPCRs and their biochemical characteristics in vitro is challenging. Both D_2_R and D_3_R are involved in regulating locomotion, reward, and addiction, but D_3_Rs appear to have a more specific role in regulating emotional and cognitive processes that involve behavioral sensitization [37]. Evidence suggests that behavioral sensitization is closely related to the plasticity of limbic neurochemical systems [193,194], which are also implicated in a range of psychiatric and substance use disorders.

The roles of D_3_R in the development of behavioral sensitization have been proposed based on various observations [195]. For instance, D_3_R antagonists can block behavioral sensitization, and both transcriptional and translational activities involved in D_3_R expression are reduced following behavioral sensitization [144,196].

Previous reports have demonstrated that D_3_R has inhibitory effects on the synthesis and release of DA in the striatum [197,198,199]. D_3_R also appears to mediate the inhibition of neuronal firing [200,201], and selective D_3_R agonists can lower extracellular levels of DA, leading to the inhibition of locomotion [202,203].

Although the signaling pathways of D_2_R and D_3_R largely overlap, the intensity of D_2_R signaling is stronger than that of D_3_R, except in the inhibition of dopamine release [22,204,205]. Therefore, the development of the desensitization of D_3_R after repeated neuronal stimulation may result in the loss of a D_3_R-mediated ‘brake’ on dopamine release, leading to increases in locomotion and behavioral sensitization [195,206].

Posttranslational modifications play critical roles in maintaining the desensitizing properties of D_2_R and D_3_R. Among the regulatory processes, such as agonist-induced receptor phosphorylation, arrestin translocation, and receptor endocytosis, only those related to intracellular trafficking occur more strongly with D_2_R than with D_3_R. Considering that receptor endocytosis is necessary for the resensitization of desensitized receptors, D_3_R is likely more susceptible to desensitization than D_2_R. In contrast, glycosylation and palmitoylation of D_3_R are necessary for maintaining its capacity to undergo desensitization.

Overall, D_3_R is predominantly expressed in limbic brain regions that are associated with various psychiatric and substance use disorders, where the plasticity of limbic neurochemical systems plays a significant role. In vitro studies have demonstrated that D_3_R undergoes dynamic desensitization that necessitates specific PTMs. As D_3_R appears to function as a negative regulator for various behavioral aspects that are linked to the dopaminergic nervous system, it is justifiable to propose a theoretical model of behavioral sensitization that views it as the desensitization of inhibitory systems, where D_3_R has a vital function.

## Figures and Tables

**Figure 1 ijms-24-06742-f001:**
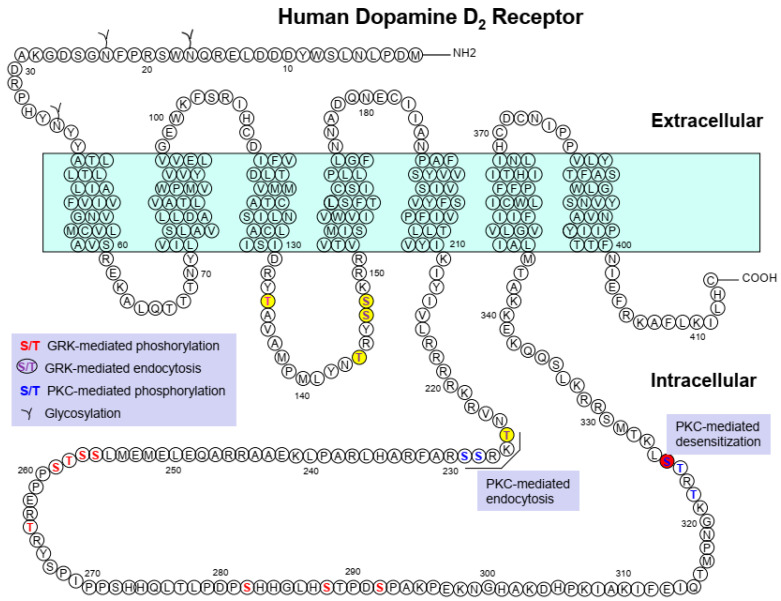
Major posttranslational modification sites of dopamine D_2_ receptor and their functional roles. S/T residues highlighted in red indicate GRK-mediated phosphorylation sites; circles filled with yellow and S/T residues highlighted in purple indicate the amino acids responsible for GRK-mediated D_2_R endocytosis; S/T residues highlighted in blue indicate PKC-mediated phosphorylation sites.

**Figure 2 ijms-24-06742-f002:**
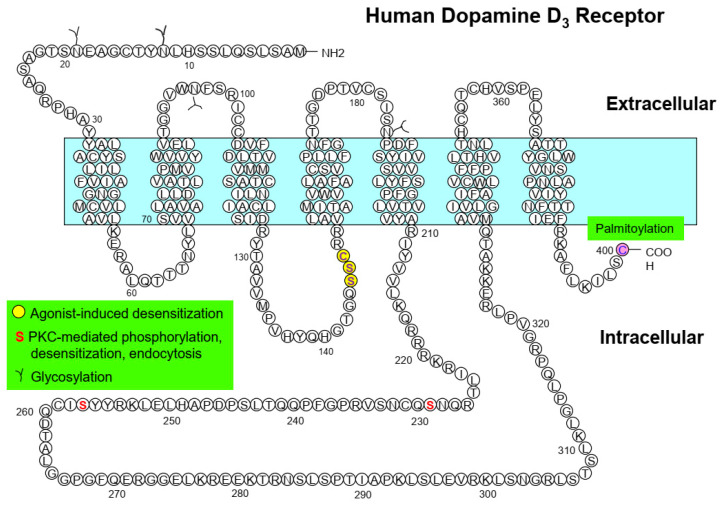
Major posttranslational modification sites of dopamine D_3_ receptor and their functional roles. Circles filled with yellow indicate the amino acids responsible for agonist-induced desensitization; the two S residues highlighted in red indicate PKC-mediated regulatory processes.

**Figure 3 ijms-24-06742-f003:**
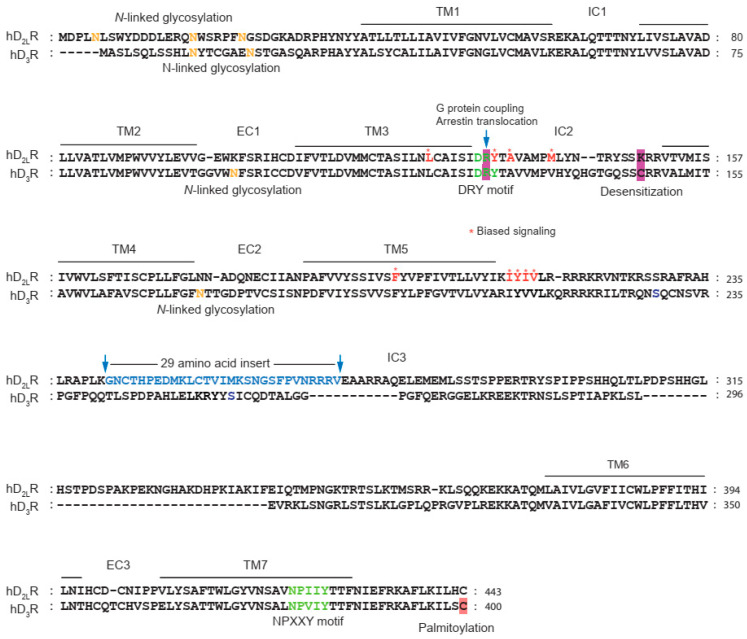
Sequence alignment of dopamine D_2_ and D_3_ receptors. D_2L_R represents alternatively spliced long form of dopamine D_2_ receptor. The asparagine residues (N) indicated by orange letters located within the N-terminal tail or within the extracellular loops represent potential *N*-linked glycosylation sites. Asterisks indicate amino acid residues involved in biased signaling. For details, please refer to the text.

**Table 1 ijms-24-06742-t001:** Roles of posttranslational modification of dopamine D_2_ and D_3_ receptors.

PTM	Receptor	Sites (Region)	Functional Roles
Glycosylation	D_2_R	N5, N17, N23 (N-terminus)	Caveolar endocytosis [57,58]
	D_3_R	N12, N19 (N-terminus)	Clathrin-mediated endocytosis [58]
		N97 (ECL1), N173 (ECL2)	Receptor endocytosis [58]
Phosphorylation	D_2_R	S256, S257, T258, S259, T264, S282, S288, S292	Agonist-induced phosphorylation, receptor recycling [51,53]
		S228, S229, T322, T324, S325	PKC-mediated phosphorylation, receptor endocytosis; desensitization (S325) [56]
	D_3_R	S229, S257	PKC-mediated phosphorylation, endocytosis, desensitization [43]
Ubiquitination	D_2_R	None	
	D_3_R	?	PMA-induced lysosomal downregulation [59]
Palmitoylation	D_2_R	None	
	D_3_R	C400	Cell-surface expression, PKC-mediated endocytosis, agonist affinity, desensitization [31]

**Table 2 ijms-24-06742-t002:** Regions involved in the biased signaling of dopamine D_2_ receptor.

Sites	Specificity (G Protein vs. Arrestin)
F198A (F^5.38^)	G protein > Arrestin [108]
R132L/L123W (R^3.50^/L^3.41^)	G protein > Arrestin [109]
^212^IYIV^215^ → AAAA	G protein > Arrestin [110]
L125N/Y133L	G protein > Arrestin [111]
A135R/M140D	G protein < Arrestin [111]

## Data Availability

Data sharing not applicable.

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
