# Peer review of "Unveiling the Differences in Signaling and Regulatory Mechanisms between Dopamine D2 and D3 Receptors and Their Impact on Behavioral Sensitization"

_ijms, 2023, doi:10.3390/ijms24076742_

Round 1

Reviewer 1 Report

Dear Authors. 

Thank You For the possibility of reviewing Your manuscript.

I have only few suggestion. The first considers the title. Maybe You should change it to be more eye-catching.

Maybe Your ought to exclude some references as their number is quite high.

The overall organisation of research seem to be proper and well prepared.

Overall the article is interesting and well prepared

Your sincerely

Author Response

Thank you for your valuable comments.
Based on Referee's comments, the title of the review has been changed to: "Unveiling the differences in signaling and regulatory mechanisms between dopamine D2 and D3 receptors, and their impact on behavioral sensitization"

Reviewer 2 Report

G-protein coupled receptors or GPCRs have been a focal point of modern biology for decades. The set of these integral membrane proteins embedded into the lipid bilayer of the cell surface structure forms one of the largest protein families in the living organism. Due to their very diverse biological functions, GPCRs represent one of the most important areas of basic research, and also applied research related to drug development.

There is no doubt, therefore, that Professor Kyeong-Man Kim literature review article entitled "Signaling and regulatory mechanisms of dopamine D2 and D3 receptors" submitted for publication may arouse the interest of many professionals. The author is an excellent connoisseur of the dopamine receptor topic, as well as a distinguished researcher.

Professor Kim's article discusses D2 and D3 receptors in detail, primarily the various signaling pathways related to them, including biased signaling / biased agonists. He also reviews post-translational modifications and regulatory mechanisms, as well as receptor dimerization, endocytosis, intracellular trafficking.

I found the text part of the work to be fine. Unfortunately, however, the presented three figures and their corresponding figure legends are completely mixed up, so this definitely needs to be improved! After correcting the figures and their texts, I recommend accepting and publishing the review paper.

„Közösség által ellenÅ‘rizve” ikon

Author Response

Thank you for your valuable comments.
According to Referee's opinion, figures and table legends were carefully checked and corrected.

Reviewer 3 Report

The review presents a thorough discussion dealing with the main signalling and regulatory mechanisms of dopamine D2 and D3 receptors. I believe the work executed by the author in reviewing and assembling all the information in this manuscript is outstanding. However, I believe the main focus on the reason why this review is needed seems a bit unclear.

1.     The author should clearly explain in the introduction what is the main aim of this review, as it is far from being understandable at the moment.

2.     What is firstly needed is a much more critical comparison with previous reviews that deal with this well-known topic.

3.     It would also important to quantitively report the number of papers dealing with dopamine D2 and D3 receptors from databases (e.g. Scopus).

4.     An article that would be important to cite in this review: https://journals.plos.org/plosone/article?id=10.1371/journal.pone.0044316

5.     Please report a 3D structure of the proteins in addition to the classical sequence alignments.

Author Response

Thank you for your valuable comments. According to Referee's opinion, the manuscript was carefully checked and corrected.

1. The author should clearly explain in the introduction what is the main aim of this review, as it is far from being understandable at the moment.

To begin with, lines 68-72 of the manuscript have clearly stated the objective of this review. Additionally, lines 668-675 were included to explicate the findings of this review, making it simpler for readers to comprehend the purpose and outcomes of this study.

2. What is firstly needed is a much more critical comparison with previous reviews that deal with this well-known topic.

As mentioned in lines 72-76 of the manuscript, several excellent reviews primarily address the in vivo roles of these receptors or the discovery of novel ligands. In contrast, this review mainly concentrates on the biochemical research domain, which sets it apart from other reviews.

3. It would also important to quantitively report the number of papers dealing with dopamine D2 and D3 receptors from databases (e.g. Scopus).

While I appreciate the valuable feedback from the referee, I have chosen not to incorporate it into my review due to various technical issues, such as the selection of only the biochemical aspect among the papers. I appreciate your understanding.

4. An article that would be important to cite in this review: https://journals.plos.org/plosone/article?id=10.1371/journal.pone.0044316

Yes, we cited it.

5. Please report a 3D structure of the proteins in addition to the classical sequence alignments.

As per the referee's recommendation, I have included a brief overview of the recently reported 3D structures in lines 404-415 of the manuscript. However, due to my lack of expertise in this area, I have refrained from providing detailed technical descriptions. Interested readers can refer to other comprehensive reviews for more in-depth information.

Round 2

Reviewer 3 Report

The author has sufficiently solved the raised issues. Now the paper can be accepted.